# Attention Guided Deep Supervision Model for Prostate Segmentation in Multisite Heterogeneous MRI Data

**Kuruparan Shanmugalingam**[1]    KURUPARAN@UNSW.EDU.AU
**Arcot Sowmya**[1]    A.SOWMYA@UNSW.EDU.AU
**Daniel Moses**[1,2]    DANIEL.MOSES@UNSW.EDU.AU
**Erik Meijering**[1]    ERIK.MEIJERING@UNSW.EDU.AU

[1] *University of New South Wales, Sydney, Australia*
[2] *Prince of Wales Hospital, Sydney, Australia*

## Abstract

Prostate cancer and benign prostatic hyperplasia are common diseases in men and require early and accurate diagnosis for optimal treatment. Standard diagnostic tests such as the prostate-specific antigen test and digital rectal examination are inconvenient. Thus, non-invasive methods such as magnetic resonance imaging (MRI) and automated image analysis are increasingly utilised to facilitate and improve prostate diagnostics. Segmentation is a vital part of the prostate image analysis pipeline, and deep neural networks are now the tool of choice to automate this task. In this work, we benchmark various deep neural networks for 3D prostate segmentation using four different publicly available datasets and one private dataset. We show that popular networks such as U-Net trained on one dataset typically generalise poorly when tested on others due to data heterogeneity. Aiming to address this issue, we propose a novel deep-learning architecture for prostate whole-gland segmentation in T2-weighted MRI images that exploits various techniques such as pyramid pooling, concurrent spatial and channel squeeze and excitation, and deep supervision. Our extensive experiments demonstrate that it performs superiorly without requiring special adaptation to any specific dataset.

**Keywords:** Prostate, Image Segmentation, Neural Networks, Deep Learning, Attention Mechanisms, Magnetic Resonance Imaging, Domain Heterogeneity.

## 1. Introduction

Prostate cancer (PCa) and benign prostatic hyperplasia (BPH) are common diseases in men (Bray et al., 2018; Rawla, 2019). Frontline tests used by clinicians for identification, screening and staging of these diseases include a prostate-specific antigen (PSA) test and digital rectal examination (DRE), however these suffer from low diagnostic performance. Multiparametric MRI has shown great promise and increased performance in diagnosing PCa and assessing BPH (Samarasinghe, 2018; Weinreb et al., 2016; Cuocolo et al., 2020). In particular, T2-weighted MRI provides good anatomical definition owing to its high spatial resolution and soft-tissue contrast, facilitating image analysis.

Image segmentation is a vital part of any automated image analysis pipeline for prostate diagnostics. Many methods have been developed for this purpose and several public datasets have been made available in the past decade to evaluate and compare their performance, notably from the MICCAI 2012 Grand Challenge on Prostate MR Image Segmentation

2012 (PROMISE12) (Litjens et al., 2014), the Initiative for Collaborative Computer Vision Benchmarking (I2CVB) (Lemaître et al., 2015), and the NCI-ISBI 2013 Challenge on Automated Segmentation of Prostate Structures (Farahani et al., 2013).

Despite significant progress reported in published benchmarks, accurate prostate segmentation on new datasets remains a challenge, with much room for further performance improvement. Virtually all recent top-performing prostate segmentation methods are based on deep learning of artificial (in particular convolutional) neural networks. However, in practice, a deep-learning model trained on a specific dataset tends to perform well only on that dataset and does not generalise well to data from other sites. Especially, a model trained on public data is not guaranteed to perform well on a private dataset. Also, in multisite studies, the data can be quite heterogeneous, due to the use of different scanners, sequences, field strengths, resolutions, matrix sizes, coils and other factors.

There have been several recent attempts to develop more generalisable deep-learning solutions for biomedical image segmentation. A notable example is nnU-Net (Isensee et al., 2021), which automatically configures itself for any dataset and segmentation task to be performed on any hardware platform, by selecting the optimal preprocessing, network architecture, training strategy and post-processing methods. It generates three U-Net (Ronneberger et al., 2015) configurations, including a two-dimensional (2D) U-Net, a three-dimensional (3D) U-Net operating at full image resolution, and a 3D U-Net cascade with the first U-Net operating on downsampled images and the second refining the segmentation maps at full resolution. After cross validation, nnU-net chooses the best-performing configuration or ensemble for the task at hand. While successful in various challenges, it is a computationally demanding framework, and it would be more practical to have a single segmentation model that performs well across heterogeneous data.

In this work, we propose a novel deep-learning architecture for prostate whole-gland segmentation in T2-weighted 3D MRI images and investigate its generalisability to multisite data from different public datasets as well as a private dataset. The architecture design exploits various techniques such as pyramid pooling, concurrent spatial and channel squeeze and excitation, and deep supervision. After a brief discussion of related work, we present the data we used, our proposed model and our experimental results and conclusions.

## 2. Related Work

Many prostate gland segmentation methods have been proposed. They may be categorised into traditional image analysis methods such as active contours (Cheng et al., 2016), conditional random fields (Toth and Madabhushi, 2012), graph-based semiautomatic methods (Artan et al., 2011), fuzzy-clustering methods (Ghose et al., 2012), and deep-learning methods such as convolutional neural networks (CNNs) (Dai et al., 2020) or fully convolutional networks (FCNs) (Long et al., 2015) including SegNet (Alqazzaz et al., 2019), PSPNet (Zhao et al., 2017), U-Net (Ronneberger et al., 2015), and others (Khan et al., 2020), or combinations of traditional and deep-learning methods (da Silva et al., 2020).

Deep-learning based segmentation methods typically use an encoder-decoder network architecture. The encoder learns the relevant image features in a latent representation, which are used by the decoder to generate the final segmentation mask. A recent evaluation of deep neural networks for prostate segmentation in T2-weighted MRI included U-Net,

SegNet, and DeeplabV3+ (Khan et al., 2020), and the current leaderboard of PROMISE12 features networks such as MSD-Net, HD-Net, and nnU-Net.

For many biomedical image segmentation tasks such as liver segmentation, lung segmentation, and brain segmentation, 2D U-Net (Ronneberger et al., 2015) and 3D U-Net (Çiçek et al., 2016) are considered to be the baseline. U-Net variants such as UNet++ (Zhou et al., 2018), RA-UNet (Jin et al., 2020), 3D $U^2$-Net (Huang et al., 2019), UNet3+ (Huang et al., 2020), Cascaded UNet (Chen et al., 2019), Attention U-Net (Oktay et al., 2018), and USE-Net (Rundo et al., 2019) have been reported to perform better on specific segmentation tasks. However, it has not been shown so far that existing models perform consistently well on prostate gland segmentation in heterogeneous data.

Starting with U-Net, several refinements have proven advantageous. Examples include the pyramid convolution block, channel attention block, and residual refinement block in HD-Net (Jia et al., 2019), the replacement of skip connections between encoder and decoder in MultiResUNet (Ibtehaz and Rahman, 2020), full-scale skip connections in UNet3+ (Huang et al., 2020), and squeeze-and-excitation blocks used successfully in a range of networks (Rundo et al., 2019; Roy et al., 2018a). In our proposed network we adapt several of them to achieve more robust prostate segmentation.

## 3. Heterogeneous Data

We collected publicly available datasets from various challenges and archives as well as private data from a local radiology practice (Table 1).

The PROMISE12 dataset contains 50 training images and their respective prostate gland annotation masks. In the challenge, a separate test set of 30 images was used for evaluation and leaderboard ranking, however their annotations are not publicly available, and therefore we used only the training dataset in our experiments. The scans are from four different institutions and were acquired using Siemens or GE scanners with or without endorectal coils and field strengths of 1.5 T or 3 T at different resolutions.

| Data | Cases | Institution | Field Strength | Resolution | Coil | Manufacturer |
|------|-------|-------------|----------------|------------|------|--------------|
| PROMISE12 | 50 | HK | 1.5 T | 0.625/3.6 | Yes | Siemens |
| | | BIDMC | 3 T | 0.25/2.2–3 | Yes | GE |
| | | UCL | 1.5 or 3 T | 0.325–0.625/3–3.6 | No | Siemens |
| | | RUNMC | 3 T | 0.5–0.75/3.6–4 | No | Siemens |
| I2CVB | 19 | HCRUDB | 3 T | 0.67/0.79 | No | Siemens |
| ISBI3T | 30 | RUNMC | 3 T | 0.6–0.625/3.6–4 | Yes | Siemens |
| ISBIPD | 30 | BMC | 1.5 T | 0.4/3 | Yes | Philips |
| Private | 50 | SMI | 3 T | 0.35/3.3 | Yes | GE |

Table 1: Datasets used in this study. See the original PROMISE12, I2CVB, and ISBI references in the introduction for more detailed information on the public data. The private dataset was provided by Spectrum Medical Imaging, Sydney, Australia. Resolution is given in millimetres in-plane/through-plane.

To make our dataset even more heterogenous than PROMISE12, we also included the I2CVB benchmark data, which contains 19 scans from another institute acquired using a Siemens scanner at 3 T field strength and a fixed resolution.

Furthermore, we added the data from the NCI-ISBI challenge, which consists of two subsets, each containing 30 scans acquired using either a Siemens scanner at 3 T or a Philips scanner at 1.5 T, from two different institutions. In fact, the two subsets are from two different data collections of The Cancer Imaging Archive (TCIA), namely the Prostate-3T collection and the Prostate Diagnosis collection, and therefore we treat them as separate datasets and refer to them as ISBI3T and ISBIPD, respectively. The task in the challenge was to segment the prostate zones, however we merged the zonal annotation masks to suit our goal of evaluating whole-gland segmentations.

Finally, a private dataset was obtained via our local collaboration, where each scan was initially annotated by three well-trained medical students using 3D Slicer, and the masks were refined and combined by an expert radiologist. Ethics approval for use of this dataset was given by the UNSW Human Research Ethics Committee.

## 4. Proposed Model

Exploiting concepts from other areas of computer vision, we developed a new U-Net-like architecture, named 3DDOSPyUSENet (Figure 1), and evaluated it for the task of whole-

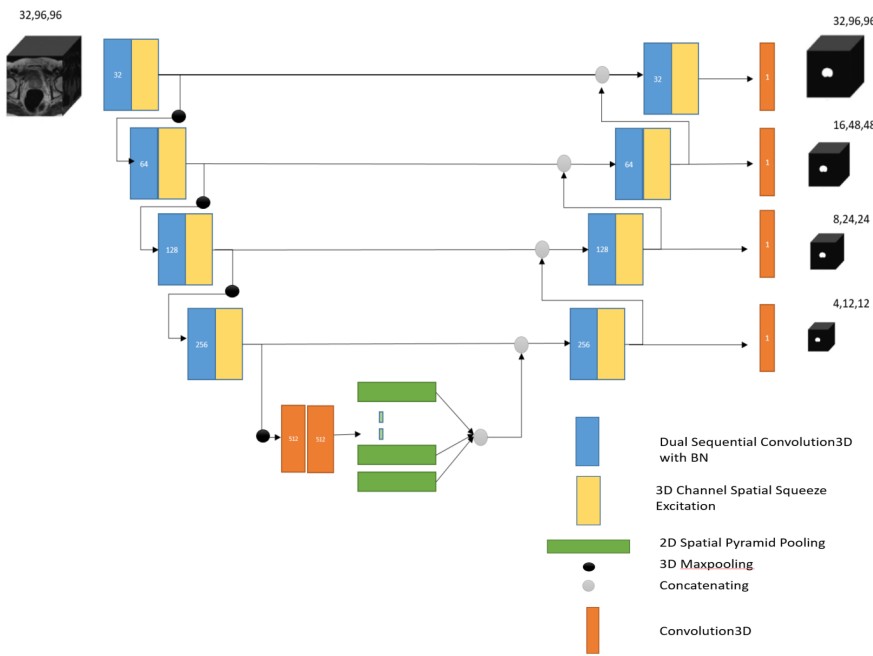

Figure 1: Proposed 3DDOSPyUSENet using deep supervision with pyramid pooling and concurrent spatial and channel squeeze and excitation.

| Model | PROMISE12 | I2CVB | ISBI3T | ISBIPD | ISBI | All | Private |
|---|---|---|---|---|---|---|---|
| 3DUNet | 0.737 | 0.562 | 0.829 | 0.726 | 0.813 | 0.439 | 0.613 |
| 3DUSENet | 0.746 | 0.500 | 0.812 | 0.767 | 0.808 | 0.399 | 0.652 |
| 3DPyUSENet | 0.791 | 0.649 | 0.805 | 0.760 | 0.823 | 0.525 | 0.669 |
| 3DDOSUNet | 0.799 | 0.791 | 0.847 | **0.786** | 0.823 | 0.755 | 0.669 |
| 3DDOSUSENet | 0.777 | 0.745 | 0.845 | 0.773 | 0.817 | 0.727 | 0.671 |
| 3DDOSPyUSENet | **0.806** | **0.792** | **0.846** | 0.777 | **0.829** | **0.768** | **0.673** |

Table 2: Performance of the six models on the seven datasets, measured as average Dice scores from 5-fold cross validation. ISBI denotes the combination of ISBI3T and ISBIPD. All denotes the combination of all public datasets. Best performance per dataset is indicated in bold. In most cases the variance was less than 1%.

gland prostate segmentation in heterogenous MRI. It uses deep supervision with pyramid pooling and squeeze-and-excitation modules to boost segmentation accuracy.

Pyramid pooling was originally used in the pyramid scene parsing network (PSPNet) architecture as an effective global contextual prior (Zhao et al., 2017). It fuses image information from four different pyramid scales which, after upsampling, are concatenated with the original feature map. The use of a pyramid pooling module has been shown to improve performance, for example in skin lesion segmentation (Shahin et al., 2019) where the module was used throughout many layers, making it computationally expensive. In our design, we use pyramid pooling only in latent space to aggregate contextual information from different regions rather than global context information.

Squeeze and excitation (SE) approaches can be categorised into channel SE (cSE), spatial SE (sSE), and concurrent channel-spatial SE (csSE). SE helps in feature map recalibration to suppress weak features and enhance meaningful features. Based on the original experiments with SE (Roy et al., 2018a), csSE performs better than either of cSE and sSE alone in most cases. Subsequent experiments using SE layers in different places within U-Net for various biomedical image segmentation tasks in brain MRI, whole-body CT, and retinal OCT scans revealed that the best option is to place them after the last convolutional layer of every encoder and decoder block (Roy et al., 2018b).

In our proposed 3DDOSPyUSENet, each encoder and decoder block uses dual sequential convolution with batch normalisation (BN) and rectified linear unit (ReLU) activation followed by csSE, and 2D pyramid pooling is applied to each channel of the 3D input in latent space (lowest level in the U-Net). Our network also uses multi-output deep supervision inspired by DeepHIPs (Manjon et al., 2020) rather than a single-output approach (Liu et al., 2020; Dou et al., 2017). To this end, the segmentation masks were downsampled to the resolution of each decoder level (Figure 1).

## 5. Experimental Results

The datasets were downloaded in NIFTI or DICOM format as provided and we used the SimpleITK library for visual inspection and data conversion. Each 3D image was resampled to a matrix of $x \times y \times z = 96 \times 96 \times 32$ voxels. As the different datasets have different slice spacing, we resampled the images to obtain the same slice spacing, and subsequently

| Training \ Testing | Model | PROMISE12 Avg | PROMISE12 Ens | I2CVB Avg | I2CVB Ens | ISBI3T Avg | ISBI3T Ens | ISBIPD Avg | ISBIPD Ens | Private Avg | Private Ens |
|---|---|---|---|---|---|---|---|---|---|---|---|
| PROMISE12 | 1 | | | 0.625 | 0.694 | 0.832 | 0.853 | 0.610 | 0.633 | 0.621 | 0.653 |
| | 2 | | | 0.671 | 0.714 | 0.858 | 0.874 | 0.624 | 0.646 | 0.648 | 0.662 |
| | 3 | | | 0.695 | **0.731** | 0.863 | 0.874 | 0.650 | 0.550 | 0.660 | 0.670 |
| | 4 | | | 0.659 | 0.677 | 0.863 | 0.877 | 0.728 | 0.739 | 0.671 | 0.678 |
| | 5 | | | 0.661 | 0.682 | 0.847 | 0.863 | 0.688 | 0.709 | 0.664 | 0.672 |
| | 6 | | | 0.639 | 0.673 | 0.861 | **0.878** | 0.725 | **0.778** | 0.670 | **0.679** |
| I2CVB | 1 | 0.086 | 0.075 | | | 0.139 | 0.127 | 0.061 | 0.066 | 0.076 | 0.103 |
| | 2 | 0.125 | 0.120 | | | 0.201 | 0.194 | 0.100 | 0.094 | 0.109 | 0.118 |
| | 3 | **0.168** | 0.165 | | | 0.347 | **0.378** | 0.110 | 0.099 | 0.178 | **0.200** |
| | 4 | 0.078 | 0.080 | | | 0.268 | 0.275 | 0.169 | 0.153 | 0.149 | 0.150 |
| | 5 | 0.050 | 0.040 | | | 0.090 | 0.051 | 0.074 | 0.050 | 0.048 | 0.021 |
| | 6 | 0.097 | 0.101 | | | 0.288 | 0.300 | 0.188 | **0.190** | 0.138 | 0.144 |
| ISBI3T | 1 | 0.374 | 0.425 | 0.357 | 0.363 | | | 0.233 | 0.206 | 0.226 | 0.249 |
| | 2 | 0.382 | 0.423 | 0.427 | 0.544 | | | 0.235 | 0.233 | 0.262 | 0.304 |
| | 3 | 0.432 | **0.579** | 0.371 | 0.440 | | | 0.202 | 0.192 | 0.258 | 0.299 |
| | 4 | 0.402 | 0.431 | 0.538 | 0.549 | | | 0.364 | 0.374 | 0.341 | 0.349 |
| | 5 | 0.349 | 0.358 | 0.573 | 0.602 | | | 0.290 | 0.278 | 0.333 | 0.341 |
| | 6 | 0.476 | 0.499 | 0.616 | **0.644** | | | 0.510 | **0.521** | 0.379 | **0.385** |
| ISBIPD | 1 | 0.080 | 0.087 | 0.203 | 0.272 | 0.362 | 0.469 | | | 0.146 | 0.150 |
| | 2 | 0.098 | 0.122 | 0.247 | 0.257 | 0.394 | 0.412 | | | 0.262 | **0.268** |
| | 3 | 0.106 | 0.120 | 0.171 | 0.135 | 0.408 | 0.439 | | | 0.180 | 0.242 |
| | 4 | 0.176 | 0.175 | 0.276 | 0.275 | 0.488 | **0.497** | | | 0.220 | 0.214 |
| | 5 | 0.085 | 0.063 | 0.256 | 0.247 | 0.439 | 0.438 | | | 0.138 | 0.124 |
| | 6 | **0.178** | 0.172 | **0.309** | 0.304 | 0.478 | 0.486 | | | 0.250 | 0.232 |
| Private | 1 | 0.412 | 0.478 | 0.492 | 0.542 | 0.482 | 0.548 | 0.471 | 0.546 | | |
| | 2 | 0.408 | 0.478 | 0.511 | 0.548 | 0.582 | 0.602 | 0.512 | 0.561 | | |
| | 3 | 0.475 | 0.498 | 0.600 | 0.659 | 0.685 | 0.711 | 0.577 | 0.622 | | |
| | 4 | 0.525 | 0.529 | 0.646 | 0.667 | 0.731 | 0.741 | 0.645 | 0.662 | | |
| | 5 | 0.484 | 0.506 | 0.666 | 0.695 | 0.727 | 0.741 | 0.656 | 0.683 | | |
| | 6 | 0.525 | **0.540** | 0.666 | **0.702** | 0.731 | **0.745** | 0.661 | **0.686** | | |

Table 3: Performance of the six models when trained on one of the five datasets and tested on the other four. Models 1–6 are 3DUNet, 3DUSENet, 3DPyUSENet, 3DDO-SUNet, 3DDOSUSENet, and 3DDOSPyUSENet, respectively. Values shown are either average (Avg) Dice scores from 5-fold cross validation or the Dice scores of ensembling (Ens) the predictions of the folds. Best performance per training-test dataset pair is indicated in bold.

normalised and saved them as NumPy arrays for model training. In each experiment presented here, we split the data 80%:20% into training and test sets respectively, and further split the former into 85%:15% for training and validation.

Using 3D U-Net (hereafter more concisely called 3DUNet) as the baseline, we evaluated the performance of our 3DDOSPyUSENet and investigated the effects of pyramid pooling (Py), squeeze and excitation (SE), and deep output supervision (DOS) by ablation. Specifically, we considered 3DUSENet (3DUNet with SE), 3DPyUSENet (with Py and SE), 3DDOSUNet (with DOS), 3DDOSUSENet (with SE and DOS). All models were trained and tested on an Nvidia V100 GPU of the Gadi server at the National Computer Infrastructure (NCI) in Australia. The Anaconda virtual environment with TensorFlow 2.1 was

used for the experiments. Stochastic gradient descent was used as optimizer with learning rate 0.001 and all models were trained for up to 150 epochs.

First we evaluated the performance of each of the six models on each of the five datasets as well as on the combined ISBI data sets and on all public datasets combined. We employed the commonly used Dice similarity coefficient (DSC) as the performance metric (Taha and Hanbury, 2015). For each of the $6 * (5 + 2) = 42$ experiments we used 5-fold cross validation and computed the averaged Dice value, increasing the total number of experiments performed to $42 * 5 = 210$. From the results (Table 2) we observe that Py, SE, and DOS had different effects. Specifically, adding SE to 3DUNet improved the performance especially on the PROMISE12 and private datasets, but it was not necessarily beneficial on the other public datasets. Adding Py on top of SE yielded either improved performance in most cases or otherwise had little negative effect. Adding DOS without Py and SE generally improved the results, whereas the combination of DOS and SE performed worse. Finally, the full model performed best overall and seems a safe choice. It produced fairly good results even for the most heterogeneous dataset (All).

Next we evaluated the performance of the six models when trained on any of the original five datasets (Table 1) and tested on any of the four others. Here again we used 5-fold cross validation, and considered the average Dice score of the folds as well as the Dice score after ensembling the predictions of the folds. Thus in total $6 * 5 * 4 * 5 = 600$ experiments were performed. From the results (Table 3) we see that the models trained on PROMISE12 performed well when tested on the I2CVB, ISBI3T, ISBIPD, and private data, whereas models trained on any of the latter generalised relatively poorly. This can be explained by the fact that the PROMISE12 dataset itself is already an heterogeneous collection of data from various sources. As in the previous experiment, we observe that Py, SE, and DOS have different effects, however their combination works well in most cases. Also, we conclude that ensembling seems to perform better than averaging. We found that when trained on all public datasets and tested on our private dataset, our proposed 3DDOSPyUSENet model had an average Dice score of 0.6517, which is similar to training on PROMISE12 and superior to the other models, from which we conclude that the other training datasets made little difference.

Although the presented results are encouraging, both quantitatively (Tables 2 and 3) and qualitatively (Figure 2), they are not yet as high as the best scores on the current PROMISE12 leaderboard. However, our results cannot be compared directly with the latter, as the test sets are different (the PROMISE12 test set is not publicly available, so we used only the training set, from which we used a subset for training and the rest for testing). In a pilot experiment (data not shown) we tested published methods such as nnU-Net on the training subset we used here and found that 3DDOSPyUSENet gave better results. Therefore we suspect that when using the full training set, rather than a subset, our method will perform comparably or better than some of those on the leaderboard.

## 6. Conclusion

Public challenges in the past decade have stimulated the development of improved methods for prostate segmentation in 3D MRI images and have shown the great potential of deep-learning based methods for this purpose. Despite success stories in individual benchmarks,

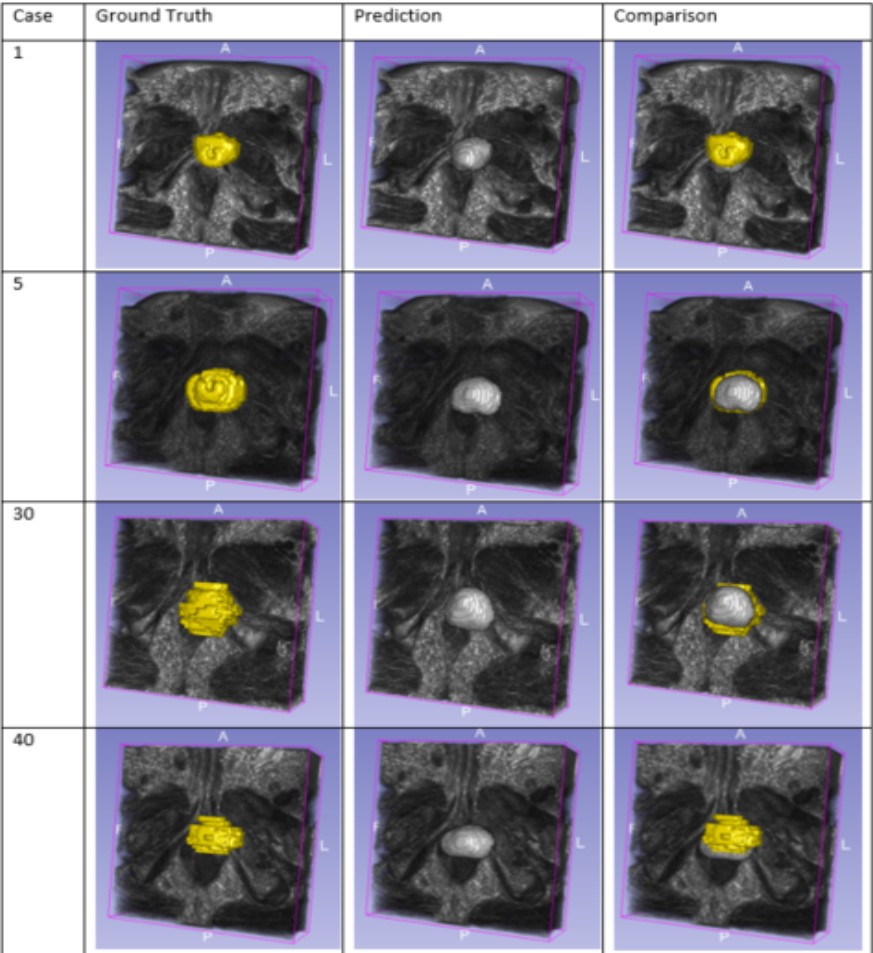

Figure 2: Visual comparison (volume renderings) of 3D segmentation results (prediction) of our proposed 3DDOSPyUSENet model and the consensus manual annotations (ground truth) for four sample cases from our private dataset.

the performance of models trained on one dataset and tested on another is often quite limited, especially on private data that may not match the characteristics of public datasets very well. In this paper we have demonstrated this to be the case with the popular baseline 3D U-Net model. We have also proposed a new model, namely 3DDOSPyUSENet, that exploits pyramid pooling, concurrent channel and spatial squeeze and excitation, and deep output supervision. Our extensive experiments clearly show the superior performance of the new model across multisite heterogeneous public and private data. From this we conclude that the use of these modifications help in suppressing weak features, enhancing meaningful features, and reducing the loss of information throughout the different layers of the network. In future work we aim to compare our new model with a wider range of existing baseline models and obtain the scores on the test sets of ongoing public benchmarks.

## Acknowledgments

The presented research was undertaken with the assistance of resources and services provided by the National Computational Infrastructure (NCI) organization which is supported by the Australian Government. Kuruparan Shanmugalingam is supported by an University International Postgraduate Award (UIPA) Scholarship funded by UNSW. The authors also gratefully acknowledge the data annotation efforts of the UNSW VIP InsightMed students in 2021.

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
