# OpenReview forum: "Attention Guided Deep Supervision Model for Prostate Segmentation in MultiSite Heterogeneous MRI Data"
_MIDL.io/2022/Conference — MIDL 2022_

### Meta-Review · Area_Chair_CAyJ · 2022-02-19

**Recommendation:** Accept (Poster)
**Confidence:** 5

**Metareview:**

The authors proposed a prostate whole-gland segmentation in T2-weighted 3D MRI images using novel deep-learning architecture, which investigates its generalisability to multisite data from different public datasets as well as a private dataset. It explores the pyramid pooling, concurrent spatial and channel squeeze and excitation, and deep supervision. Various recent innovations in a novel model are demonstrated experimentally to better generalize across datasets in prostate image analysis. The experimental results demonstrate the proposed method is promising. Building a more generalizable model the extensive evaluation results are a practical contribution to the field.

---

### Decision · Program_Chairs · 2022-02-28

Accept